# Fundamentals, Diagnostic Capabilities, and Perspective of Narrow Band Imaging for Early Gastric Cancer

**DOI:** 10.3390/jcm10132918

**Published:** 2021-06-29

**Authors:** Hiroki Kurumi, Kouichi Nonaka, Yuichiro Ikebuchi, Akira Yoshida, Koichiro Kawaguchi, Kazuo Yashima, Hajime Isomoto

**Affiliations:** 1Division of Gastroenterology and Nephrology, Department of Multidisciplinary Internal Medicine, Faculty of Medicine, Tottori University, 36-1 Nishicho, Yonago 683-8504, Japan; kurumi_1022_1107@yahoo.co.jp (H.K.); ikebu@tottori-u.ac.jp (Y.I.); akirayoshida1021@yahoo.co.jp (A.Y.); koichiro@tottori-u.ac.jp (K.K.); yashima@tottori-u.ac.jp (K.Y.); 2Department of Digestive Endoscopy, Tokyo Women’s Medical University Hospital, 8-1 Kawada-cho, Shinjuku-ku, Tokyo 162-8666, Japan; nonaka513@gmail.com

**Keywords:** narrow band imaging, gastric cancer, image-enhanced endoscopy, texture and color enhancement imaging, red dichromatic imaging

## Abstract

The development of image-enhanced endoscopy has dramatically improved the qualitative and quantitative diagnosis of gastrointestinal tumors. In particular, narrow band imaging (NBI) has been widely accepted by endoscopists around the world in their daily practice. In 2009, Yao et al. proposed vessel plus surface (VS) classification, a diagnostic algorithm for early gastric cancer using magnifying endoscopy with NBI (ME-NBI), and in 2016, Muto et al. proposed a magnifying endoscopy simple diagnostic algorithm for early gastric cancer (MESDA-G) based on VS classification. In addition, the usefulness of ME-NBI in the differential diagnosis of gastric cancer from gastritis, diagnosis of lesion extent, inference of histopathological type, and diagnosis of depth has also been investigated. In this paper, we narrative review the basic principles, current status, and future prospects of NBI.

## 1. Introduction

According to the latest cancer statistics, 1.09 million new cases of stomach cancer are diagnosed annually worldwide, and 770,000 people die of stomach cancer annually. It is the sixth most common cancer and the third most common cause of death among all cancer types [1]. Gastric cancer has a 5-year survival rate of over 95% if detected early; therefore, early detection is pivotal for better prognosis [2]. To this end, endoscopic diagnosis of gastric tumors has improved with the development and advancement of endoscopic equipment.

In particular, the development of image-enhanced endoscopy, represented by narrow band imaging (NBI), has dramatically improved the qualitative and quantitative diagnosis of gastrointestinal tumors. Narrow-band light with peaks at 415 nm and 540 nm, which is used in NBI, is shorter wavelength of visible light and has low tissue permeability, making it ideal for observing mucosal surface structures [3]. These two short wavelengths coincide with the peak absorption region of oxidized hemoglobin, and mucosal capillaries are observed as a clear low signal compared to the surrounding tissue. The evaluation of intrapapillary capillary loops within the esophageal epithelium was established for the qualitative and quantitative diagnosis of esophageal tumors [4,5,6]. Although dye spraying has been used as a conventional adjunct diagnosis to white light imaging (WLI), it is no exaggeration to say that NBI has made it possible to avoid the risk of iodine allergy in the examinee and physical burden such as heartburn, as in the case of iodine staining of the esophagus [7,8]. On the other hand, in NBI observation of mucosa with glandular structures, the light projected onto the marginal crypt epithelium (MCE) does not reach the blood vessels and causes backward confusion, which is visualized as a white border. This is called the white zone (WZ), and the morphology of the glandular structure can be estimated [9]. Furthermore, when used in combination with magnifying endoscopy, the mucosal capillaries and glandular structures can be evaluated in more detail (Figure 1). While various categories of image-enhanced endoscopy have been applied in clinical practice, NBI, when used in combination with magnifying endoscopy, is particularly good at observing fine anatomical structures and is expected to be useful in the endoscopic diagnosis of gastric cancer. In this paper, we review the basic principles, current status, and future prospects of magnifying endoscopy with NBI (ME-NBI).

## 2. Basic Observation Methods and Points of ME-NBI for Gastric Cancer

### 2.1. ME-NBI for Gastric Non-Cancerous Mucosa

Since the normal gastric mucosa shows a wide variety of mucosal patterns due to differences in glandular areas and inflammatory modifications, knowing these typical images is extremely important in ME-NBI diagnosis of gastric cancer.

In the gastric fundic gland mucosa without inflammation, the gland ducts are arranged regularly in a concave position perpendicular to the mucosal surface. When this is evaluated by ME-NBI, the crypt openings are observed as brown dots, and the surrounding MCE as a WZ. Capillaries of the intervening part around the WZ have a brown mesh pattern (Figure 2a). Inflammatory changes caused by *Helicobacter pylori* (*HP*) infection gradually distort the gland ducts. The crypt openings become oval or indistinct, and the WZ, which is circular, changes to oval shape (Figure 2b). As the inflammatory changes progressed, the oval WZ gradually become tubular and the capillaries of the intervening part are indistinct or coil-shaped (Figure 2c). Then, when the gastric fundic gland disappears, the appearance resembles the mucosa of the pyloric gland (Figure 2d). The pyloric gland mucosa shows little change in the ME-NBI image due to inflammation. In the pyloric gland mucosa without inflammation, the gland duct is distorted, and the intervening part protrudes into the mucosal surface in a ridged pattern. When it was observed by ME-NBI, MCE appears as a regular tubular WZ, and coil-shaped capillaries are observed in the intervening part (Figure 2e) [10,11].

ME-NBI of intestinal metaplasia shows a blue-white border at the limbus of the WZ. This was reported as a light blue crest by Uedo et al. and is thought to represent the difference in reflectance characteristics caused by the projection of 415 nm narrow-band light onto the brush border of intestinal metaplasia (Figure 3) [12].

### 2.2. Diagnostic Algorithm for Early Gastric Cancer Using ME-NBI

In 2009, Yao et al. proposed vessel plus surface (VS) classification, a diagnostic algorithm for early gastric cancer using ME-NBI [10], and in 2016, Muto et al. proposed a magnifying endoscopy simple diagnostic algorithm for early gastric cancer (MESDA-G) based on VS classification [11]. In MESDA-G, (1) the presence of a demarcation line (DL) is checked, and if present, (2) microvascular pattern (MVP) or microsurface pattern (MSP) will be analyzed, and if either or both are irregular, early gastric cancer is diagnosed. The anatomical structures used in MVP analysis are subepithelial capillaries, collecting venules, and pathological microvessels. A pathological microvessel is a general term for small blood vessels that cannot be classified as subepithelial capillaries or collecting venules. The anatomical structures used in the analysis of MSP are the WZ, which reflects the MCE, crypt openings, intervening part, and white opaque substance (WOS) (Figure 4). WOS is a white substance that exists in the surface of intestinal metaplasia, adenoma, and carcinoma of chronic gastritis mucosa reported by Yao et al. WOS is a microscopic lipid droplets that accumulates in the epithelium and subepithelium. The morphology and distribution of WOS can be used to differentiate between cancer and non-cancerous lesions. The presence of WOS also suggests gastric or gastrointestinal mucin phenotype [13,14,15,16].

MVP morphology can be classified as regular, irregular, or absent. Regular MVPs have a uniform morphology and shape for each vessel, symmetrical distribution, and regular arrangement. Irregular MVP has a variety of morphologies, including looped, tortuous, branched, and bizarrely shaped vessels with asymmetric distribution and irregular arrangement. Cancer-specific morphology of irregular micro vessels has been described as dilatation, heterogeneity in shape, abrupt caliber alteration, and tortuousness. Absent MVP has WOS on the mucosal surface, and MVP cannot be adequately observed.

MSP morphology is classified as regular, irregular, or absent. In regular MSP, WZ is a uniform linear, curved, oval, or circular structure with homogeneous morphology, symmetrical distribution, and regular arrangement. If the WOS is present and the MSP is not visible, the MSP is regular if the WOS is regularly arranged in a uniform form. In irregular MSP, the morphology of WZs is an irregular linear, curved, oval, circular, or villous structure with heterogeneous morphology, asymmetrical distribution, and irregular arrangement. When WOS is present, the morphology of WOS is non-uniform and the arrangement is irregular; this is called irregular MSP. Absent MSP is the state in which the WZ and WOS are not visible (Figure 5, Table 1) [11].

On the other hand, ME-NBI findings specific to early gastric cancer have also been reported, including the presence of white globular reservoirs less than 1 mm in size beneath the intraepithelial vessels, termed white globe appearance (WGA) by Doyama et al. (Figure 6). WGA reflects an intraglandular necrotic debris, which has been reported as a specific histopathological marker for cancer, and is present in 20% of early gastric cancers, but not in adenomas, and has a high specificity for early gastric cancer [17].

### 2.3. Inference of Histopathological Type of Gastric Cancer by ME-NBI

Tubular adenocarcinomas proliferate while retaining the morphology of the glandular structure. In well-differentiated adenocarcinomas with relatively straight and vertically concave gland ducts, the capillaries in the intervening part are in a reticular pattern, termed the mesh pattern by Yagi et al. (Figure 7). Capillaries in the mesh pattern observed in early gastric cancer show dilatation, caliber change, and uneven shape, and can be differentiated from gastritis. Well-differentiated adenocarcinomas with a mesh pattern tend to have dense glandular structures, and the WZ is often not visible, so the DL can be clearly visualized. DL is also useful for differentiating gastritis from gastric cancer. A regular mesh pattern is called a complete mesh pattern and is suggestive of a well-differentiated mucosal adenocarcinoma [18]. This pattern is considered to be synonymous with the fine network pattern reported by Nakayoshi et al. [19]. In tubular adenocarcinomas, where the gland ducts are tortuous and branched, the WZ is irregularly tubular, and the capillaries in the intervening part are irregularly looped (Figure 8). This is referred to as the loop pattern proposed by Yagi et al. [18]. In addition, Kanemitu et al. reported the characteristic findings of papillary adenocarcinoma visualized using ME-NBI, with vessels within the circular intervening part surrounded by circular MCE, which labeled as the vessels within the epithelial circle (VEC) pattern. They reported that 94.3% of early gastric cancers with a VEC pattern showed papillary structures on histopathology, and the frequency of submucosal invasion was significantly high in early gastric cancers with a VEC pattern [20]. Papillary adenocarcinoma is more biologically aggressive than tubular adenocarcinoma and requires more attention [21,22,23]. 

The ME-NBI findings in undifferentiated carcinomas, including poorly differentiated adenocarcinomas and signet-ring cell carcinomas, vary depending on the degree of invasion. In the early stage, non-cancerous glandular ducts remain in the superficial layer and the number of cancer cells is small; therefore, ME-NBI often shows only subtle changes, such as an enlarged intervening part. With further invasion, poorly connected, shrunken microvessels can be observed. Furthermore, when undifferentiated adenocarcinoma invaded all layers of the mucosa and the glandular structure disappeared, the WZ disappeared, the density of blood vessels decreased, and microvessels in the form of contracted surfaces, each of which meanders irregularly, were observed (Figure 9). This is referred to as wavy microvessels by Yagi et al. [18]. However, the majority of early undifferentiated adenocarcinomas are covered by a non-cancerous epithelium, and the ME-NBI image is poorly altered, limiting the diagnosis. Capture of slight changes like enlargement of the intervening part warrants biopsy evaluation.

### 2.4. Efficacy of ME-NBI for Early Gastric Cancer

#### 2.4.1. ME-NBI in Screening Tests for Gastric Cancer

Yao et al. reported that 97% of early gastric cancers met the VS classification criteria, and Ezoe et al. reported that the accuracy rate of VS classification for small gastric depression cancer was 96.6% [24]. In a multicenter prospective study, Yao et al. reported the usefulness of ME-NBI using VS classification in routine examinations. Especially in erythematous/isochromatic mucosal lesions, mainly differentiated adenocarcinoma, the accuracy rate was extremely high at 99.4%, and I think that NBI is an essential technique for routine screening examinations outside of referral centers [25]. 

Marta et al. performed a meta-analysis to evaluate the diagnostic value of NBI for gastric intestinal metaplasia and early gastric cancer. The pooled sensitivity and specificity of NBI for gastric intestinal metaplasia were 0.79 and 0.91, respectively. The tubulovillous pattern was the most accurate marker for detecting gastric intestinal metaplasia and could be evaluated effectively without the need for high magnification. The pooled sensitivity and specificity of NBI for gastric cancer were 0.87 and 0.97, respectively. The use of magnification improved the performance of NBI in characterizing gastric cancer, especially when VS classification was applied [26].

These reports suggest that NBI is useful in screening tests and is an essential technique in daily practice.

#### 2.4.2. ME-NBI in Histological Diagnosis of Gastric Cancer

Nonaka et al. reported the usefulness of ME-NBI in differentiating gastric adenomas from well-differentiated adenocarcinomas. Type I is when the WZ is preserved and relatively uniform with distinct MVP; Type II, when the WZ is relatively uniform and the MVP is similar to the surrounding mucosa; Type III is when the WZ is relatively uniform with irregular, darker MVP that is more prominent than the surrounding mucosa; Type IV is when the WZ tends to disappear with presence of irregular MVP; and Type V is when the WZ disappears with presence of irregular MVP. The accuracy rate was 88.5%, with type I and II as adenomas and type III to V as well-differentiated adenocarcinomas [27]. 

Yao et al. reported that 100% of the lesions with well-formed WOS were adenomas, and 83% of the lesions with irregular WOS were well-differentiated adenocarcinomas [15].

The usefulness of ME-NBI for the histopathological diagnosis of gastric cancer requires further investigation in the future.

#### 2.4.3. ME-NBI in Determining the Horizontal Extent of Gastric Cancer

Nonaka et al. examined the concordance rate between DL and histopathologic borders observed in depressed and flat-differentiated adenocarcinomas. The accuracy rate was 100%, indicating that ME-NBI is useful for diagnosing the extent of differentiated adenocarcinoma [28]. Especially in cases where the irregularity of the WZ is clear, or in lesions where the WZ is not visible and the irregularity of the MVP is easily visible, determining the horizontal extent is easy. However, some lesions, such as extremely well-differentiated adenocarcinoma, have only minor irregularities in the WZ and MVP; therefore, caution is required. 

Horii et al. examined the rate of negative biopsies from non-cancerous tissue outside the lesion and negative horizontal margins of endoscopic submucosal dissection specimens in early gastric cancer after confirmation of DL using ME-NBI.

The rates of biopsy-negative and negative horizontal margins were 96.7% and 97.9%, respectively, in early gastric cancer. They reported that the risk factors for a positive horizontal resection margin were tumor size > 20 mm and moderately or poorly differentiated adenocarcinomas [29].

Horiuchi et al. examined the concordance rate between the DL and histopathological borders in undifferentiated adenocarcinomas. They reported a concordance rate of 81.6%, which was 27.6% higher than that of white light observation alone. In particular, undifferentiated adenocarcinoma with little inflammatory cell infiltration could be delineated with high accuracy [30]. However, undifferentiated adenocarcinoma may develop laterally with non-cancerous epithelium, and ME-NBI has limitations in accurately diagnosing its extent. In such cases, a biopsy should be performed in combination with ME-NBI findings to make a decision.

In recent years, the number of gastric cancers detected after *HP* eradication has been increasing, and it has been reported that gastric cancers detected after *HP* eradication may have a gastritis-like surface structure, which may make qualitative diagnosis and determining extent difficult. [31]. Saka et al. reported that the histopathological reason for this was that non-cancerous epithelium and non-cancerous glandular ducts covered the cancerous glandular ducts [32].

Akazawa et al. reported that there were significantly more gastric cancers with unclear DL in the *HP* eradication group than in the *HP* infection group (11.8% vs. 1.5%) [33].

Horiguchi et al. investigated the accuracy of determining the horizontal extent of gastric cancer after *HP* eradication and reported that WLI, chromoendoscopy, and ME-NBI all significantly decreased the reliability of the determining the horizontal extent of gastric cancer after *HP* eradication compared with non-eradication [34].

The number of gastric cancers detected after *HP* eradication is expected to increase in the future, and further studies on the margin delineation for gastric cancer detected after eradication are needed.

#### 2.4.4. ME-NBI in Estimating the Depth of Invasion of Early Gastric Cancer

Kikuchi et al. reported that the presence of dilated abnormal blood vessels (three times larger than microvessels) in early gastric cancer may correlate with submucosal invasion, but the number of cases was small and the sensitivity was 37.5%, which is not sufficient [35]. Yagi et al. reported that 94.9% of early gastric cancers with a mesh pattern were intramucosal carcinomas and 92.3% of early gastric cancers with an interrupted pattern were submucosal invasive carcinomas, but the number of cases was small, and further studies are needed [18]. 

The usefulness of ME-NBI in estimating the invasion depth has been reported in recent years, but no definite opinion has been obtained, and this is a subject for further study. 

## 3. Comparison of ME-NBI with Other Modalities

Blue laser imaging (BLI) is an observation method that is often compared with NBI. BLI uses two short-wavelength narrow-band lasers of 410 nm and 450 nm. The 410 nm laser mainly highlights the surface blood vessels and structures of the mucosa. The output of the 450 nm laser light was adjusted to excite the phosphor to produce light with a wide wavelength to ensure brightness. Depending on the balance of output power, BLI can be used in two modes: BLI mode, which is suitable for magnification, and BLI-bright mode, which is slightly brighter and suitable for non-magnified observation [36].

Le et al. compared the diagnostic accuracy for gastric cancer of conventional WLI, magnifying endoscopy with WLI (ME-WLI), ME-NBI, and magnifying endoscopy with BLI (ME-BLI) in a meta-analysis including eight prospective studies. They reported that the diagnostic accuracy of ME-WLI, ME-NBI, and ME-BLI was higher than that of conventional WLI. They also reported that the diagnostic accuracy of ME-NBI and ME-BLI was significantly higher than that of ME-WLI, but there was no difference between ME-NBI and ME-BLI [37].

Zhou et al. compared the diagnostic efficacy of NBI and BLI in detecting gastric cancer in a meta-analysis of six BLI and 22 NBI reports. The pooled sensitivity of BLI for gastric cancer was 0.89, and the specificity was 0.92. The pooled sensitivity of NBI for gastric cancer was 0.83 and the specificity was 0.95. There was no difference in the diagnostic performance of NBI and BLI, and both groups had high diagnostic performance [38].

The results of the two meta-analyses showed no significant difference between the diagnostic performance of NBI and BLI.

## 4. Future Prospects

Till date, the fundamentals and diagnostic capabilities of NBI for early gastric cancer are outlined. Although ME-NBI has become an essential technique for screening and precision examination of gastric cancer at an earlier stage, it is somewhat complicated, requires a higher level of expertise, and is still subjective. Today, artificial intelligence (AI) has great potential to support decision making in various medical fields, and may be able to detect abnormalities that are often overlooked by non-experts. Li et al. developed a new system based on convolutional neural network to analyze the early gastric cancer observed by ME-NBI. The results showed that the sensitivity, specificity, and accuracy in diagnosing early gastric cancer were 91.18%, 90.64%, and 90.91%, respectively. In addition, there was no significant difference in the specificity and accuracy of diagnosis between their system and experts. Moreover, the diagnostic sensitivity, specificity, and accuracy of their system were significantly higher than those of the non-experts [39]. Hu et al. developed a computer-aided diagnostic model for early gastric cancer to analyze and assist in the diagnosis of early gastric cancer using ME-NBI. They compared the model with eight endoscopists with varying experience, and found that their model achieved similar predictive performance to the senior endoscopists (accuracy: 0.770 vs. 0.755, *p* = 0.355; sensitivity: 0.792 vs 0.767, *p* = 0.183; specificity: 0.745 vs. 0.742, *p* = 0.931) but better than the junior endoscopists (accuracy: 0.770 vs. 0.728, *p* < 0.05). After referring to the results, the average diagnostic ability of the endoscopists was significantly improved in terms of accuracy, sensitivity, positive and negative predictive value [40]. 

These results suggest that the combination of ME-NBI and artificial intelligence (AI) is useful for the diagnosis of early gastric cancer. The combination of AI has the potential to overcome the weaknesses of NBI in that it is subjective and requires a high level of expertise, suggesting that it can diagnose non-experts at the same level as experts, and research in this direction is expected to further enhance the diagnostic capabilities of NBI.

In addition to NBI, a variety of other observation methods have emerged. For the detection of tumors, observation methods that emphasize color tone and structure, such as linked color imaging, have been developed and are widely used, and there are reports on their usefulness [41]. In addition, the development of texture and color enhancement imaging (TXI), which is a new observation method that emphasizes color tone and structure, extended depth of field, which is easy to focus, and red dichromatic imaging (RDI), which is a new narrow-band light observation method that improves the visibility of deep tissues such as deep blood vessels and improves the visibility of bleeding points, has been reported. TXI is designed to enhance three image factors in WLI (texture, brightness, and color) to define subtle tissue difference clearly. TXI has two settings, “mode1” with color enhancement and “mode2” without color enhancement. TXI mode2 looks more like WLI color tone (Figure 10) [42]. RDI is a new image-enhanced endoscopy consisting of three types of illumination with wavelengths of 540, 600, and 630 nm. Hemoglobin strongly absorbs light at 600 nm and weakly absorbs light at 630 nm. Because of this absorption property, the light intensity of the reflected light at 600 nm is greatly attenuated compared to that of the reflected light at 630 nm in areas with high hemoglobin concentration, such as the bleeding point, and it is observed as an orange color. On the other hand, when the hemoglobin concentration is low around the bleeding point, the light intensity of the reflected light at 600 nm and 630 nm is about the same, and it is observed as yellow (Figure 11) [43]. 

In this way, the development and improvement of endoscopic equipment continues at a rapid pace, and it is important to understand the principles and advantages of each observation method before performing an examination.

## 5. Conclusions

The current status of ME-NBI for early gastric cancer was outlined. ME-NBI has become an essential procedure for screening and detailed examination, but it is somewhat complicated and subjective than that for esophagus and colon. In addition, with the increase of gastric cancer detected after HP eradication, the qualitative and extent delinetion of ME-NBI may become more complicated in the future. Although the application of AI may solve this problem, it is necessary to constantly feed back the consistency of ME-NBI and histopathological findings observed in clinical practice and to train oneself in order to improve ME-NBI techniques.

## Figures and Tables

**Figure 1 jcm-10-02918-f001:**
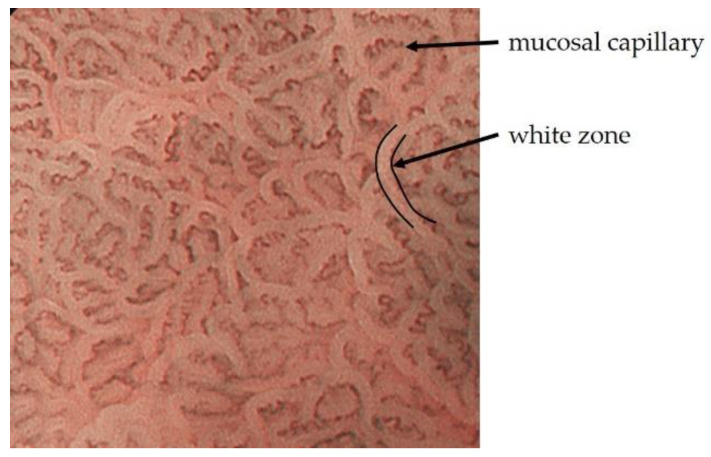
Mucosal capillaries are observed as low signal compared to the surrounding tissue. WZ reflects the morphology of the marginal crypt epithelium.

**Figure 2 jcm-10-02918-f002:**
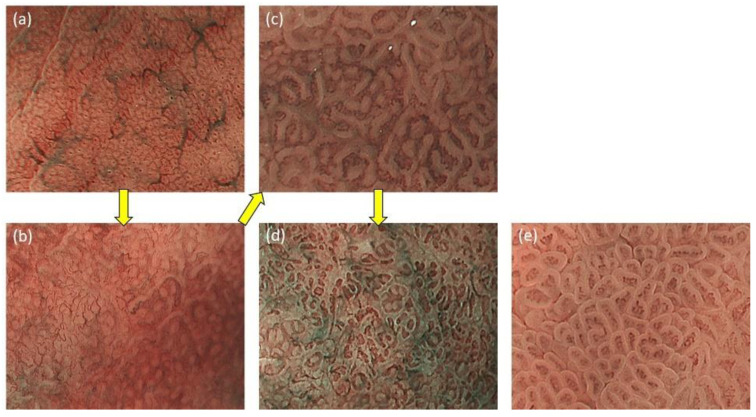
(**a**) The fundic gland mucosa without inflammation. The glandular structures are regularly arranged in a mesh pattern; (**b**) The glandular structure is still circular, but gradually becomes irregular in size and shape; (**c**) The circular glandular structures gradually disappear and tubular glandular structures are observed; (**d**) Gastric fundic glands disappear and glandular structure resembles that of pyloric glands; (**e**) The pyloric gland mucosa without inflammation.

**Figure 3 jcm-10-02918-f003:**
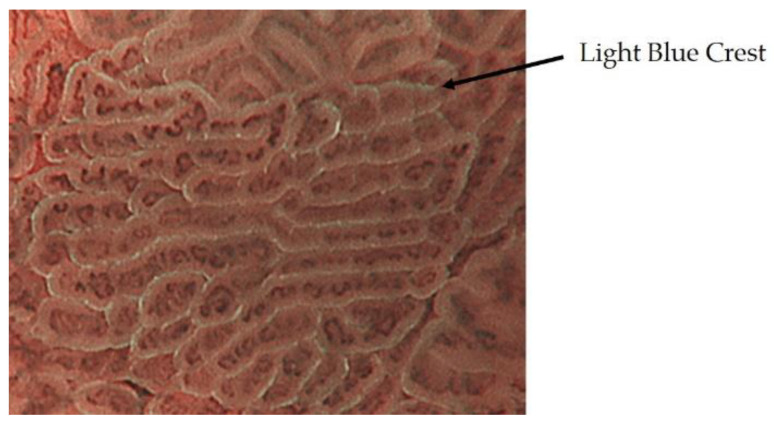
The intestinal metaplasia observed as a blue-white border by ME-NBI.

**Figure 4 jcm-10-02918-f004:**
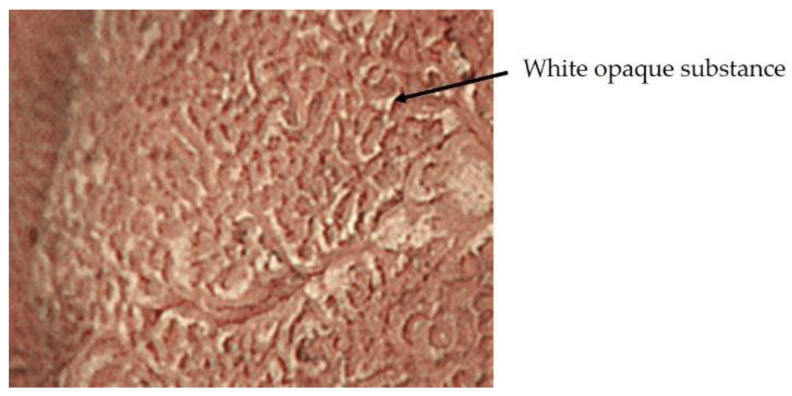
White opaque substance.

**Figure 5 jcm-10-02918-f005:**
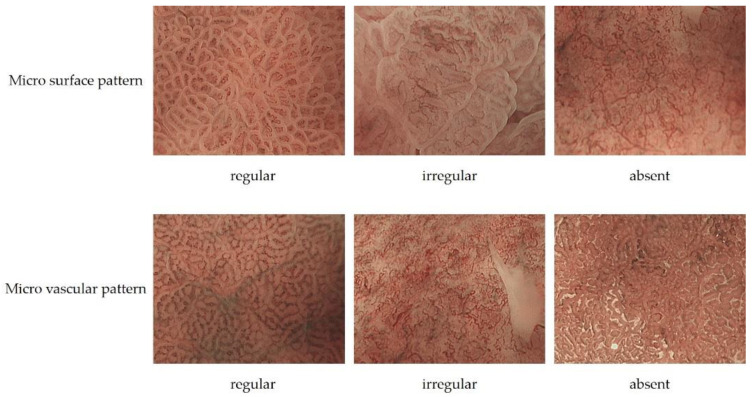
The upper row is an example of regular MSP, irregular MSP, and absent MSP. The bottom row is an example of regular MVP, irregular MVP, and absent MVP.

**Figure 6 jcm-10-02918-f006:**
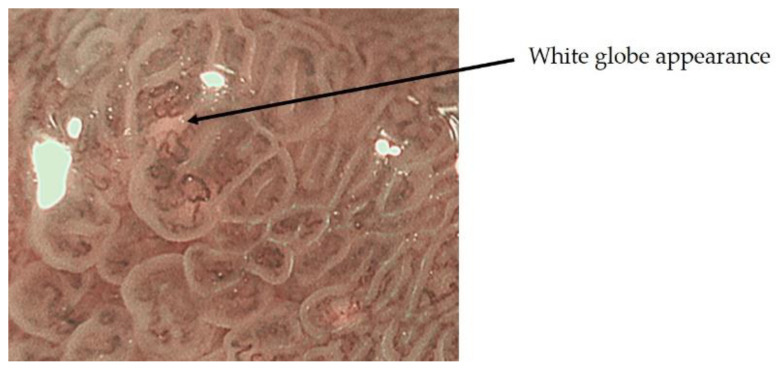
White globe appearance.

**Figure 7 jcm-10-02918-f007:**
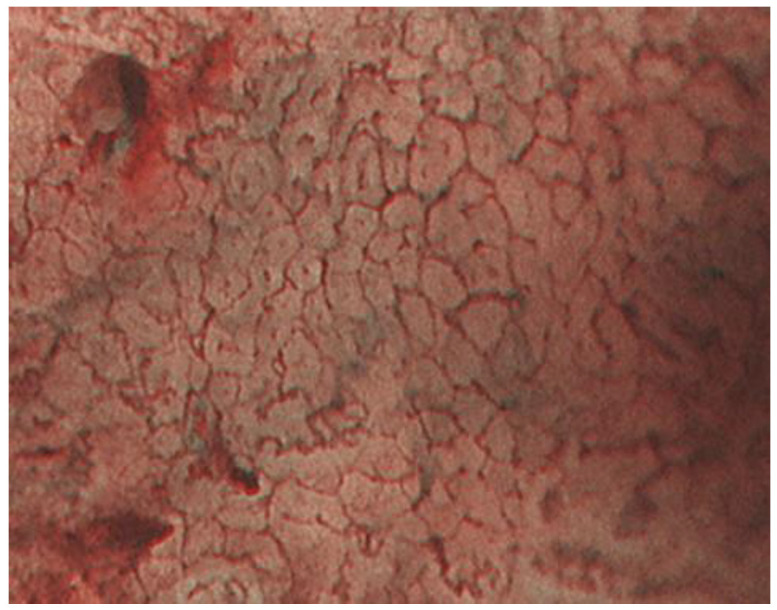
The mesh pattern suggests tubular adenocarcinomas.

**Figure 8 jcm-10-02918-f008:**
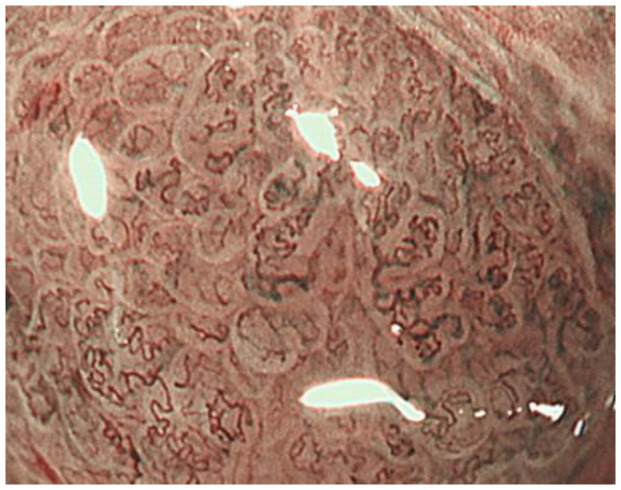
The loop pattern suggests tubular adenocarcinomas.

**Figure 9 jcm-10-02918-f009:**
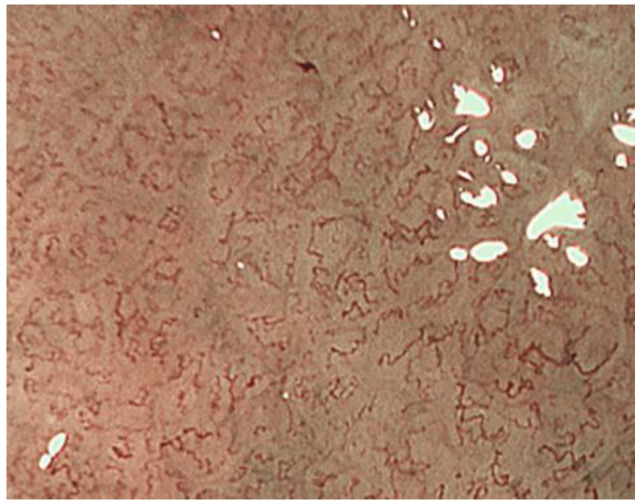
No glandular structures were observed, and wavy micro vessels were observed, suggesting undifferentiated adenocarcinoma.

**Figure 10 jcm-10-02918-f010:**
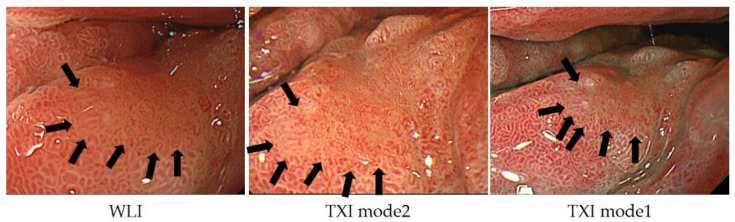
In TXI mode2, the structure is enhanced compared to WLI, and the relatively dark areas in the back are adjusted to be brighter; in TXI mode 1, the color tone is further enhanced and the contrast is more distinct.

**Figure 11 jcm-10-02918-f011:**
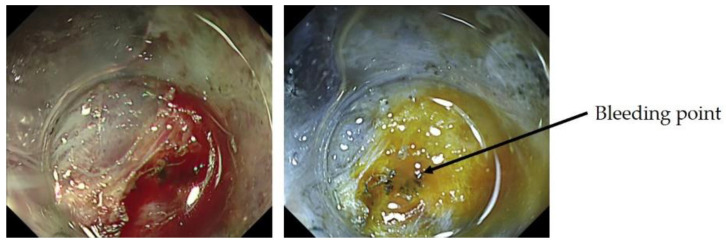
No glandular structures were observed, and wavy micro vessels were observed, suggesting undifferentiated adenocarcinoma.

**Table 1 jcm-10-02918-t001:** Differences in MSP and MVP between cancer and non-cancer.

	Non Cancer	Cancer
Micro surface pattern	morphology: homogeneous (uniform linear, curved, oval, circular) distribution: symmetric arrangement: regular	morphology: heterogeneous (irregular linear, curved, oval, circular) distribution: asymmetric arrangement: irregular
Micro vascular pattern	morphology: uniform distribution: symmetric arrangement: regular	morphology: heterogeneous (dilatation, heterogeneity in shape, abrupt caliber alteration, tortuousness) distribution: asymmetric arrangement: irregular

## Data Availability

No new data were created or analyzed in this study. Data sharing is not applicable to this article.

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
