# Peer review of "Fundamentals, Diagnostic Capabilities, and Perspective of Narrow Band Imaging for Early Gastric Cancer"

_jcm, 2021, doi:10.3390/jcm10132918_

Round 1
Reviewer 1 Report
This was a review of the efficacy of image-enhanced endoscopy, especially narrow band imaging, in the diagnosis of EGC. This manuscript is quite well written, and explains the characteristic endoscopic findings in an easy-to-understand manner.
There are some minor errors, and some points need to be improved.
1. Line 30: The epidemiology of gastric cancer needs to be updated according to the most recent data available (e.g. Globocan 2020).
2. Line 101: Please explain more about 'white opaque substance'. What is the morphological characteristics or definition of WOS? Describe the clinical meaning of WOS.
3. Line 226: Does the expression 'range diagnosis' mean prediction of tumor margin (=margin delineation) ? The expression seems somewhat unfamiliar. Please consider changing the expression.
Author Response
Thank you very much for your suggestions. We have addressed your comments, and we feel that the manuscript has now been greatly improved as a result. Please see the revised manuscript and confirm our corrections.
- Line 30: The epidemiology of gastric cancer needs to be updated according to the most recent data available (e.g. Globocan 2020).
Thank you very much for the valuable comment.
As you suggested, we have revised the manuscript from 2018 statistics to 2020 statistics.
→According to the latest cancer statistics, 1.09 million new cases of stomach cancer are diagnosed annually worldwide, and 770,000 people die of stomach cancer annually. It is the sixth most common cancer and the third most common cause of death among all cancer types [1].
- Line 101: Please explain more about 'white opaque substance'. What is the morphological characteristics or definition of WOS? Describe the clinical meaning of WOS.
Thank you very much for the valuable comment.
The noted information is very important, so we have added the following text.
→WOS is a white substance that exists in the surface of intestinal metaplasia, adenoma, and carcinoma of chronic gastritis mucosa reported by Yao et al. WOS is a microscopic lipid droplets that accumulates in the epithelium and subepithelium. The morphology and distribution of WOS can be used to differentiate between cancer and non-cancerous lesions. The presence of WOS also suggests gastric or gastrointestinal mucin phenotype [13-16].
- Line 226: Does the expression 'range diagnosis' mean prediction of tumor margin (=margin delineation) ? The expression seems somewhat unfamiliar. Please consider changing the expression.
Thank you very much for the valuable comment.
As you suggested, we have revised the manuscript.

Reviewer 2 Report
Sound evaluation and diagnostics. Discussing more in depth the briefly cited benefits of AI algorithms would have added a new perspective to the conventional approach.
Author Response
Thank you very much for your suggestions. We have addressed your comments, and we feel that the manuscript has now been greatly improved as a result. Please see the revised manuscript and confirm our corrections.
Sound evaluation and diagnostics. Discussing more in depth the briefly cited benefits of AI algorithms would have added a new perspective to the conventional approach.
Thank you very much for the valuable comment.
As you suggested, we have revised the manuscript and added the following text.
→Today, artificial intelligence (AI) has great potential to support decision making in various medical fields, and may be able to detect abnormalities that are often overlooked by non-experts. Li et al. developed a new system based on convolutional neural network to analyze the early gastric cancer observed by ME-NBI. The results showed that the sensitivity, specificity, and accuracy in diagnosing early gastric cancer were 91.18%, 90.64%, and 90.91%, respectively. In addition, there was no significant difference in the specificity and accuracy of diagnosis between their system and experts. Moreover, the diagnostic sensitivity, specificity, and accuracy of their system were significantly higher than those of the non-experts [40].

Reviewer 3 Report
Dear authors,
The current review paper presents the fundamentals of NBI for early gastric cancer interestingly. The structure of the review has been well designed and supported by evidence from available data. However, I want to suggest some changes.
- in my opinion, the manuscript is overloaded by the number of abbreviations. I would limit this. It may be confusing for the readers
- I would re-reorganize Figure 2 and present it as a path of changes
- showing in the table the differences in NBI features between non-cancerous and cancerous changes would be an added value
- Please consider adding a short section or a few sentences about using it in daily practice outside the referral centers, appropriate competency level achievement, and training
- summary/ conclusion should be more specific; please consider adding bullet points there, may be in points
Author Response
Thank you very much for your suggestions. We have addressed your comments, and we feel that the manuscript has now been greatly improved as a result. Please see the revised manuscript and confirm our corrections.
- in my opinion, the manuscript is overloaded by the number of abbreviations. I would limit this. It may be confusing for the readers
Thank you very much for your valuable comments. As you suggested, we have reduced the number of abbreviations. Specifically, we have removed EGCM, IP, and ME.
- I would re-reorganize Figure 2 and present it as a path of changes
Thank you very much for your valuable comments. I have modified Figure 2 as you suggested.
- showing in the table the differences in NBI features between non-cancerous and cancerous changes would be an added value
Thank you very much for your valuable comments. We have added Table1 as you suggested.
- Please consider adding a short section or a few sentences about using it in daily practice outside the referral centers, appropriate competency level achievement, and training
Thank you very much for your valuable comments. We have added the following text as you suggested.
→In a multicenter prospective study, Yao et al. reported the usefulness of ME-NBI using VS classification in routine examinations. Especially in erythematous/isochromatic mucosal lesions, mainly differentiated adenocarcinoma, the accuracy rate was extremely high at 99.4%, and I think that NBI is an essential technique for routine screening examinations outside of referral centers [25].
- summary/ conclusion should be more specific; please consider adding bullet points there, may be in points
Thank you very much for your valuable comments, I have created a section on Conclusion and added the following text.
→The current status of ME-NBI for early gastric cancer was outlined. ME-NBI has become an essential procedure for screening and detailed examination, but it is somewhat complicated and subjective than that for esophagus and colon. In addition, with the increase of gastric cancer detected after HP eradication, the qualitative and extent delinetion of ME-NBI may become more complicated in the future. Although the application of AI may solve this problem, it is necessary to constantly feed back the consistency of ME-NBI and histopathological findings observed in clinical practice and to train oneself in order to improve ME-NBI techniques.
